# Application of a Statistical and Linear Response Theory to Multi-Ion Na^+^ Conduction in NaChBac

**DOI:** 10.3390/e23020249

**Published:** 2021-02-21

**Authors:** William A. T. Gibby, Olena A. Fedorenko, Carlo Guardiani, Miraslau L. Barabash, Thomas Mumby, Stephen K. Roberts, Dmitry G. Luchinsky, Peter V. E. McClintock

**Affiliations:** 1Department of Physics, Lancaster University, Lancaster LA1 4YB, UK; carlo.guardiani@uniroma1.it (C.G.); m.barabash@lancaster.ac.uk (M.L.B.); thomas.mumby@googlemail.com (T.M.); d.luchinsky@lancaster.ac.uk (D.G.L.); 2School of Life Sciences, University of Nottingham, Nottingham NG7 2UH, UK; olena.fedorenko@nottingham.ac.uk; 3Division of Biomedical and Life Sciences, Lancaster University, Lancaster LA1 4YQ, UK; s.k.roberts@lancaster.ac.uk; 4Department of Mechanical and Aerospace Engineering, Sapienza University, 00185 Rome, Italy; 5KBR Inc., Ames Research Center, Mountain View, CA 94035, USA

**Keywords:** ion channel, statistical theory, linear response, ionic transport, NaChBac

## Abstract

Biological ion channels are fundamental to maintaining life. In this manuscript we apply our recently developed statistical and linear response theory to investigate Na${}^{+}$ conduction through the prokaryotic Na${}^{+}$ channel NaChBac. This work is extended theoretically by the derivation of ionic conductivity and current in an electrochemical gradient, thus enabling us to compare to a range of whole-cell data sets performed on this channel. Furthermore, we also compare the magnitudes of the currents and populations at each binding site to previously published single-channel recordings and molecular dynamics simulations respectively. In doing so, we find excellent agreement between theory and data, with predicted energy barriers at each of the four binding sites of ∼4,2.9,3.6, and 4kT.

## 1. Introduction

Biological channels are natural nanopores that passively transport ions across cellular membranes. These channels are of enormous physiological and pharmacological importance, and so investigation of their transport properties is an area of great interest and research. For example, Na${}^{+}$ channels play a key role in the generation of the action potential [1,2,3]. Furthermore, artificial nanopores are primarily designed for their transport functionality which can be informed by our understanding of biological channels.

A primary function of these channels is their ability to discriminate effectively between ions, whilst still conducting them at high rates. An example is NaChBac from *Bacillus halodurans*, which is the first bacterial voltage-gated sodium channel (Nav) to have been characterised, and thus is a prokaryotic prototype for investigating the structure–function relationship of Nav channels [4]. It conducts ions at rates of 10${}^{7}$ s${}^{-1}$ despite having permeability ratios favouring Na${}^{+}$ over K${}^{+}$ and over Ca${}^{++}$. Recently we reported these values to be *at least* 10:1 and 5:1 respectively [5]. In fact from the reversal potential the Na${}^{+}$/K${}^{+}$ permeability ratio is found to be 25:1, which is closer in agreement but still less than [6] who found the ratio to be 170:1. This contrasts with potassium channels such as KcsA where selectivity is reversed, favouring K${}^{+}$ over Na${}^{+}$ at 1000:1 [7]. The channel itself is formed from several coupled subsystems, but we focus on the selectivity filter (SF), which is the primary region responsible for selectivity between ions. The SF can readily be mutated to generate a range of conducting (and non-conducting) channel types which exhibit different selectivity and conductivity properties compared to those exhibited by the wild-type (WT) channel (see [5]).

The SF has the amino acid (Here: T = threonine, L = Lecucine, E = glutamte, S = serine, W = tryptophan, A = alanine and x highlights where the sequence is not conserved and can be several possible amino acids.) sequence TLESWAS, and thus shares the TxExW sequence with eukaryotic calcium channels [6]. Unfortunately, a crystal structure of NaChBac is not available. However, Guardiani et al. [8,9,10] applied homology structural modelling to produce a structure of NaChBac that we will use in this publication. We conduct a variety of different Molecular Dynamics (MD) simulations (see Figure 1) to explore its properties. During simulation the SF was found to have an average radius $R_{c}∼$ 2.8 Å, length $L_{c}∼$ 12 Å and 4 binding sites for conducting Na${}^{+}$ ions labelled S1–4 from the intra- to the extra-cellular side respectively. The conduction mechanism was found to involve knock-on between at least two, if not three, ions. Each binding site has a volume, as estimated in Table 1, whose sum gives the total volume of the pore $V_{c}$. The first two sites are formed at the backbone carbonyls of the threonine and leucine residues respectively. S1 is wider than the average pore radius with diameter 3.06 Å, but S2 has the average pore radius of 2.8 Å. As a result, these two sites accommodate the primary hydration shell with around 5–6 waters per ion, and thus prevent bare ion-protein interaction. S3 is of approximately the same size as S2, but the ion only interacts with four waters because it also interacts directly with the glutamate ring. The fourth site is formed on the extracellular side from the side chain of the serine residues and a sodium ion here has a 40% probability of interacting with one or two serines and a 60% probability of being fully hydrated by water. This is in stark contrast to the narrower potassium channels where K${}^{+}$ ions are almost fully dehydrated as they permeate the pore. The Na${}^{+}$ occupancies at each site have been determined by molecular simulation using 0.5 M bulk solutions. Both S1 and S4 have energy minima that are higher in energy than S2,3 and so are less likely to be occupied. In fact the average occupancy of S1,4 is only around half that of the most occupied site S2 (see Figure 7c).

These results are consistent with the results of MD simulations that have been performed on a variety of similar bacterial NaV channels. Chakrabarti et al. [12] conducted a 21.6 μs-long MD simulation of NavAb, observing a variable number of ions in the pore, mainly two or three (rarely four) and spontaneous and reversible ionic diffusion along the pore axis. Ulmschneider et al. [13] simulated the open state of the pore domain of NavMs with a voltage applied, and calculated the conductance which at ∼33pS was in agreement with experimental results.

The SF has a nominal charge of −4e arising from the fixed gluatamte ring. However, determining the exact charge contribution from these pores is challenging due to the potential partial charges from remaining uncharged amino acids and the protonation that may occur at physiological pH levels. That latter is suspected to be true in voltage-gated Ca${}^{++}$ channels which share a ring of glutamates [14,15]. As a result, protonation of the glutamate ring in Navs has been studied fairly extensively [5,16,17,18,19]. Corry and Thomas [17] investigated the pore when only a single glutamate residue was protonated. The slightly protonated pore showed little difference in the potential of mean force vs. the normal pore. However, the doubly-protonated state showed a larger barrier for permeation to the pore, and reduced affinity for ion binding. Boiteux et al. [18] found a slight difference in the average number of Na${}^{+}$ ions in the SF at 2.3 and 2.0 in the fully deprotanated and slightly-protonated states, respectively; however, both states were conducting. In simulations with two protonated residues, the authors observed the existence of a non-conducting state forming as a result of stable hydrogen bonds between the glutamates. As the number of protonated residues increased to three and four, Chloride Cl${}^{-}$ ions started to bind and the pore became non-conductive for Na${}^{+}$. A similar study with shorter biased simulations suggested that protonation of a single Glu residue would diminish the conductance [16]. Meanwhile, a recent [19] study found that, at physiological pH, the pore may exist in the full deprotonation state but that it could also exist in the single or double-protonation states as well. Furthermore, the calculated pKa value decreases with each additional bound ion, implying that the presence of ions inside the pore leads to protonation of the SF. Thus, in [5] we introduced the notion of an effective charge describing the total charge in the pore as felt by the conducting ion, and its values were estimated by fitting Brownian dynamics simulations to experimental data for wild-type (WT) NaChBac and for a large selection of mutants. In our earlier work we studied NaChBac and its mutants theoretically and by Brownian dynamics simulation [5,20].

In earlier publications [5,21], we reported studies of Na${}^{+}$ and Ca${}^{++}$ permeation in NaChBac, using Brownian dynamics models. The key result of modelling was that ionic conduction is analogous to electron transport in a quantum dot. As a function of the value of fixed charge, we observed a set of resonant conduction peaks separated by regions of blockade where the ions could not enter/leave the pore. This phenomenon is called *ionic Coulomb blockade* (ICB) [22], by analogy with (electronic) Coulomb blockade in quantum dots, for which the physics and the governing equations are essentially the same. Each resonant peak corresponds to an n→n+1 barrier-less transition, which is of the knock-on kind when n>0 [23], and the regions of blockade are when the charge carrier cannot pass. The occurrence of ICB has also been confirmed experimentally in artificial nanopores [24,25]. Although the ICB model explained immediately the role of the fixed charge, and accounted convincingly for the effect of mutations in which the fixed charge is altered, it is only a good approximation when electrostatic forces are dominant, that is, for divalent and trivalent ions. Furthermore, it does not contain affinities in the pore or excess chemical potentials in the bulk and so it cannot describe selectivity between ions of the same charge. It is also not connected to the results of Molecular simulation (MD) or the structure, and it cannot describe the absolute magnitude of the permeating current.

To provide a more accurate description, we needed a more fundamental model. We therefore developed a kinetic model [20], to investigate Na${}^{+}$ vs. K${}^{+}$ selectivity. This model was based on a simplified two site model of NaChBac and it was made self-consistent through the form of its transition rates. These were chosen such that the kinetic model and an earlier statistical and linear response theory had the same form of conductivity at low voltages. However, this did not include the complete structure or any comparison to results from MD simulation. It also did not include the binding site conductivities, or account for the correlations between ions at different binding sites. These two properties are expected to be important for fully describing the permeation properties and making quantitative predictions of the function of biological channels because it is known that small mutations in structure can lead to significant changes in function, for example, [5,26,27]. This was shown in [28], where we introduced a statistical and linear response theory fully accounting for structure and the properties of each binding site, and used it to analyse a point mutation in KcsA exploring the reasoning behind its drop in conductivity and occupancy.

In the present paper, we apply this recently developed statistical and linear response theory [28] to NaChBac with a more accurate model based on the structure introduced in [8]. The theory will include all four binding sites and their estimated volumes and surface areas, and the excess chemical potentials at each site. Furthermore, we extend this theory by deriving the conductivity at linear response in the presence of an electrochemical gradient. The theory is successfully compared to experimental single-channel and whole-cell recordings (some of which published in [5,20]), and results from MD simulations [8]. Finally, the theory allows us to make quantitative predictions of the current-concentration and current-voltage relations, and the effective open probability of the channel; as a function of the energy profile, experimental bulk concentration and structure of the pore.

In what follows, with SI units *e* is the unit charge, *T* the temperature, *z* the ionic valence, and *k* Boltzmann’s constant.

## 2. Experimental Methods and Data

To apply the theory to NaChBac, and to compare with experimental recordings and make predictions, we consider two experiments. For further details of the experimental methods, including generation of the mutant channels and their expression, as well as details of the electro-physiological experiments, we refer to [5], and here we only present a concise summary. The first of these data sets is single-channel current-voltage recordings originally published in [20]. In these experiments identical bath and pipette solutions containing (in mM: 137 NaCl, 10 HEPES and 10 glucose, pH 7.4 adjusted with 3.6 mM NaOH) were used. Single-channel recordings are possible because Na${}^{+}$ is the preferred substrate with sufficiently high conductance to provide a single-channel current amplitude which significantly exceeded noise (i.e., a favorable signal-to-noise ratio). In Figure 2a we plot the current-voltage curve, and in (b,c) we provide a current-time trace made at +100 mV. Trace (c) begins at the end of trace (b). There are at least three active channels passing currents with the magnitudes shown by the dashed lines.

In the second series of experiments, we performed whole-cell current measurements through NaChBac, in different Na${}^{+}$/K${}^{+}$ concentrations (see Figure 3). The black and purple curves in (a) (and the curve in (c)), that is, with 0M and 0.14M of NaCl solutions in the bath solution respectively (or 0.1M and 0M of KCl), were published in [5]. An identical experiment on a mutant was performed and described in [20]. In each case, the pipette solution contained (in mM) 120 Cs-methanesulfonate, 20 Na-gluconate, 5 CsCl, 10 EGTA, and 20 HEPES, pH 7.4 adjusted with 1.8 CsOH, meanwhile the bath solution contained (in mM); 137 NaCl, 10 HEPES and 10 glucose, pH 7.4 (adjusted with 3.6 mM NaOH). Permeability to K${}^{+}$ was investigated by incrementally replacing the NaCl bath solution with an equivalent KCl solution such that the total ionic concentration was fixed at 140 mM. Total current across the cell was then normalized and, because one can assume that the total number of channels, their type and their open probability is conserved in each cell for the duration of the recording, it can effectively be modeled as a single channel. This normalization was with respect to the absolute value of peak current and is shown in Figure 3a. In (b) we show the current-concentration behaviour at −10 mV, which corresponds to the peak current. The reversal potential is plotted in (c); in cases where inward current was not detected, estimated values were determined from the voltage at which outward current could be detected. Finally, in (d) and (e) we provide the corresponding current-time traces.

Since NaChBac is highly impermeable to K${}^{+}$ and Cl${}^{-}$ we have neglected the presence of these ions in the pore and in our theory we shall simply consider a single ion species, that is, Na${}^{+}$ inside the pore.

### Comparison of NaChBac Structures

In this subsection we shall compare the structure of NaChBac from the homology model which was used in [8], and the Cryo-EM structures 6vx3.pdb and 6vwx.pdb from [4].

In Figure 4 we provide an overlay of the homology model (yellow ribbons) and the 6vx3.pdb structure (green ribbons), using all of the backbone atoms. (a) provides the overlay of the whole pore and (b) provides a snap-shot of the selectivity filter (SF). From visual inspection there is clearly good agreement between the structures. In the pore the root-mean-square distance between structures (computed using the backbone atoms) is 17.47 Å and 7.14 Å in the SF.

To further explore these structures we considered the pore radius which can be compared using the HOLE program. In Figure 5 we show a comparison between structures. The homology model is more open than the Cryo-EM structures (6vx3 and 6vwx) both at the level of the cytosolic mouth (minimum centered on z=−15 Å) and in the region of the SF (around *z* = 0–12 Å). This is confirmed by volume filling representations of the pores which show a bottleneck close to the cytosolic mouth of 6vx3. The SF of 6vwx is narrower because the SF is occupied by two Na${}^{+}$ ions, and these attract the side chains of the glutamates and the backbone carbonyls of the leucines, moving them towards the centre of the pore. Hence, there are two distinct minima in the pore radius which cannot be spotted in the radius profile of the homology model because this structure was obviously empty. However, the fact that the SF in 6vx3 (whose SF is empty) is also narrower than that of the model suggests that the structural differences might reflect different functional states in the channel cycle. In fact in the paper [4], Gao comments on the narrow radius of the cytosolic mouth, and on the arrangement of the Voltage Sensor Domain, suggesting that these structures might represent an inactivated conformation of the pore. By contrast, our homology model was built using the fully open conformation of NavMs from *Magnetococcus sp.* (PDB ID: 4F4L) as a template. As a result, our homology model probably represents an open conformation of NaChBac. This choice was deliberately taken on the assumption that an open conformation would be more suitable for the computational study of permeation and selectivity. In summary, the good agreement in overlayed structures, along with the choice to use an open conformation of NavMs as a template, makes us confident our model is a reliable system for the study of the selectivity and permeation of NaChBac.

## 3. Theory

To model the SF we consider a system comprised of a pore thermally and diffusively coupled at either entrance to bulk reservoirs. This system and the effective grand canonical ensemble was considered and rigorously derived for multi-ion species in [28], and here we only present the necessary details needed to describe a single-species system. This pore is represented as a 1-dimensional lattice with 4 sites that may be occupied by a single ion at most. These are labelled S1–4 starting from the intracellular side in (c) of Figure 1. This figure also provides in (a) an overview of the system and (b) a snapshot of the SF which is highlighted by the red ribbons in (a). Clearly each configuration of Na${}^{+}$ ions in the pore represent a distinct state of the system with total state space $\left\{n_{j}\right\}$. In this system ions inside the pore interact electrostatically with each other and charges on the surface of the pore via E. Furthermore, they also interact locally at each binding site, *m*, via short-range contributions ${\bar{\mu }}_{m}^{c}$ and may experience an applied potential $\phi_{m}^{c}$. Thus, with only Na${}^{+}$ in the pore we can write the following distribution function, $P(\left\{n_{j}\right\})$,
(1)$$\begin{array}{cc} \; & P\left(\left\{n_{j}\right\}\right)=Z^{-1}\frac{{\left(x_{Na}^{b}\right)}^{n_{Na}}}{n_{0}!n_{Na}!}\exp\left[-\left(E\left(\left\{n_{j}\right\}\right)-\sum_{m}n_{Nam}\left(\Delta {\bar{\mu }}_{Nam}+ez\Delta \phi_{m}^{b}\right)\right)/kT\right]. \end{array}$$

We have introduced Δ to denote the difference between bulk and site *m* in the pore such that $\Delta {\bar{\mu }}_{m}^{b}={\bar{\mu }}^{b}-{\bar{\mu }}_{m}^{c}$ and $\Delta \phi_{m}^{b}=\phi^{b}-\phi_{m}^{c}$. In these cases $\bar{\mu }$ and ϕ denote the excess chemical potential and applied voltage in the bulk or at site *m* respectively. The prefactor contains factorial terms due to the indistinguishably of ions $n_{Na}$ and empty sites $n_{0}$ in the pore, and $x_{Na}$ denotes the mole fraction. For clarity we will drop the Na subscript. The necessary statistical properties such as site or pore occupancy can be derived from the partition function Z or Grand potential Ω=−kTlog(Z).

In [28] we demonstrated that the response to an applied electric field can be calculated following Kubo and Zwanzig [29,30,31]. We showed that the susceptibility density at each site can easily be derived and related to the conductivity at each site following the Generalised Einstein relation. The total conductivity through the pore is thus calculated by summing the reciprocals of the site-conductivity, in analogy to resistors in series. As a result all sites must be conducting for the total conductivity to be non-negligible. This effect partly explains the reduced conduction of a KcsA mutant [26], although we have to be mindful that the overall pore charge also decreases, increasing the overall energy barrier for conduction, and contributing to the reduced conductivity. We shall extend this derivation here by considering the response to an electrochemical gradient comprised of an electric potential gradient δϕ and a concentration gradient δc. We shall assume that both bulk reservoirs are perturbed symmetrically so that the left (+) and right (−) electrochemical potentials, $\mu^{b}$, can be written,
(2)$$\mu^{b}=kT\log\left(\left(c\pm \delta c/2\right)/c_{w}\right)+{\bar{\mu }}^{0}+ez\phi^{0}\pm ez\delta \phi /2,$$
where $c_{w}$ is the concentration of the solvent which is much larger than that of the ions at around ∼55M, and *c* is the concentration of the solute, ${\bar{\mu }}^{0}$ is the equilibrium bulk excess potential which we assume to be unperturbed by the electrochemical gradient and $\phi^{0}$ is the equilibrium electrical potential (which we will consider to be 0). In the following derivation we will write $c/c_{w}$ as the mole fraction *x*. Thus following [28] we can write the following free energy, $G(\left\{n_{j}\right\},\delta \phi ,\delta c)$, in the presence of this gradient by linearising $\mu^{b}$ about small δc,
(3)$$\begin{array}{ll} G(\left\{n_{j}\right\},\delta \phi ,\delta c) & =E\left(\left\{n_{j}\right\}\right)-\sum_{m=1}^{M}n_{m}\left(kT\log\left(x\right)+\Delta {\bar{\mu }}_{m}^{0}\pm \frac{kT}{2c}\delta c\pm ez\nu_{m}^{b}\delta \phi \right)\\ & +kT\mathrm{In}(n_{0})!+kT\mathrm{In}n!. \end{array}$$
In this expression we have rewritten $\delta \phi_{m}^{b}=\nu_{m}^{b}\delta \phi$ where $\nu_{m}^{b}$ is a function representing the fraction of the voltage drop to move from either the left or right bulk to site *m* in the pore (see [28] for details). In a symmetrically distributed pore (which we assume), the average of $\nu_{m}^{b}$ is equal to 1/2. In this regime the probability distribution function can be written as
(4)$$P\left(\left\{n_{j}\right\},\delta \phi ,\delta c\right)=Z^{-1}\frac{x^{n}}{n_{0}!n!}\exp[-\left(E-\sum_{m}n_{m}\left(\Delta {\bar{\mu }}_{m}^{0}\pm ez\nu_{m}^{b}\delta \phi \pm \frac{kT}{2c}\delta c\right)/kT\right].$$
Here the partition function *Z* is defined in the standard manner from the conservation of probability and distinguished from the equilibrium partition function Z. Both the free energy and distribution function can also be expressed in terms of the chemical gradient $\eta^{L}-\eta^{R}$ because
(5)$$kT\log\left(x^{L}/x^{R}\right)=\delta \eta =\frac{kT}{c}\delta c.$$
The distribution (4) can be linearised about both small δϕ and δc. When calculating the average particle density at each site ${\langle n_{m}\rangle }_{\delta c,\delta \phi }/V_{m}$, where $V_{m}$ is the site volume, one can obtain relations for the susceptibilities due to the electrical gradient $\chi_{m}^{\delta \phi }$ and the chemical or concentration gradient $\chi_{m}^{\delta \eta }$. The former is defined in [28], since we assume a symmetrical pore the latter is defined as,
(6)$$\begin{array}{c} \chi_{m}^{\delta \eta }=\frac{1}{2kT}\left(\langle n_{m}\left(\sum_{m}n_{m}\right)\rangle -\langle \left(\sum_{m}n_{m}\right)\rangle \langle n_{m}\rangle \right)\frac{1}{V_{m}}. \end{array}$$
It is worth noting that this expression is similar to $\chi_{m}^{\delta \phi }$ and is proportional to the variance of particle number at site *m* plus the covariance between sites *m* and the remaining sites in the pore. These susceptibilities are also proportional to the electrical conductivity, $\sigma_{m}$, at each binding site, which can be defined from the Einstein relation as: $\sigma_{m}=ze^{2}D_{m}\chi_{m}$ where $D_{m}$ and $\chi_{m}$ correspond to the diffusivity and susceptibility at each site respectively. As a result, the total current across the pore can be calculated as [28]
(7)$$I={\left(\sum_{m}\frac{1}{\frac{A_{m}}{L_{m}}\sigma_{m}}\right)}^{-1}\left(\delta \phi +\delta \eta /e\right),$$
where we recall that δϕ is the voltage gradient in *V*, δη is the chemical gradient in kT, and $A_{m}$ and $L_{m}$ are the surface area and length of site *m* respectively. Finally, the conductivity at each site is calculated from
(8)$$\sigma_{m}=ze^{2}D_{m}\left(\chi_{m}^{\delta \phi }+\chi_{m}^{\delta x}\right),$$
which is a function of the equilibrium bulk chemical potential.

## 4. Application to NaChBac

In Figure 6a, we consider the free energy spectra for selected (most favoured) pore configurations of NaChBac calculated from Equation (3) (when δϕ=0 and δc=0). We consider 0.14M NaCl solutions, and 0–3 ions inside the pore. In Equation (3) the total electrostatic energy, E, is calculated by approximating the pore as a capacitor of total charge $n_{f}$ and capacitance *C* taking the form $E=U_{c}{\left(n_{f}+n\right)}^{2}$ where $U_{c}=\frac{e^{2}}{2C}$ [21,22]. Since the permitivitiy of water inside the pore is not known (though it must be less than the bulk value of 80) we consider $U_{c}=10kT$. This approximation is discussed in detail in [28]. The energy spectra are parabolic vs. $n_{f}$, and each *n*-ion state has multiple configurations (15 in total) and we only highlight the most favoured. These states are determined by the values of $\Delta {\bar{\mu }}_{Nam}$, and their exact values are determined from fitting to experimental data (see Section 4.1). Differences in this term lead to energy splitting between possible configurations because the site occupied, in addition to the total number of ions inside the pore, determines the energy, *conducting* states correspond to the degeneracies where the lowest energy levels intersect, cf. [23], and this was shown to be the case in KcsA [28]. In NaChBac, the circle highlighting the 2–3 resonant transition occurs at around $n_{f}∼-2.7$. Importantly, this differs from $n_{f}=-2.5$, suggesting that the the 3rd-ion faces an energy barrier to enter each site. If the concentration of the solutions was increased the energy barrier would decrease and the location of the resonant conduction would shift along the abscissa towards $n_{f}-2.5$. It is worth reiterating that $n_{f}$ here represents the total pore charge, and so differences from the fixed glutamate ring charge of −4e can be explained from the additional contribution of all other charges and possible protonation inside the pore. Extended discussions of this point are provided in [5,16,17,18,19].

In Figure 6b we plot the energy spectra of the favoured 2 and 3-ion states, vs. $n_{f}$ but also vs. bulk concentration. From the explanation above it is clear that the latter affects the value of $n_{f}$ at which the two energy levels intersect. At low concentrations the energy barrier to add an ion to the pore is large. Thus, strong negative pore charge is required to reduce the barrier to attract the ion. Conversely at large concentrations the barrier is small and so less negative charge is needed. Thus one would expect the experimental current to be larger for measurements at higher concentrations, if these could be made.

To obtain the values of $\Delta {\bar{\mu }}_{Na,1-4}$ we performed fitting to two data sets, and this will be explained in the following subsection.

### 4.1. Comparison to Single Channel Data and MD

The values of $\Delta {\bar{\mu }}_{Na,m}$ used in Figure 6 are obtained by fitting, performed using the LSQCURVEFIT function in Matlab. We fit theory to the equilibrium site occupancies $\langle n_{Na,m}\rangle$ calculated from simulation data [8] (see Figure 2c), and the current at 35 mV. Current is needed here so that we can ensure it is of the correct order of magnitude. We also note that the difference in bulk NaCl concentration between the current and occupancy data is taken into account during fitting. To minimise the number of free parameters we also assumed that the diffusivity in the pore was constant, and equal to a tenth of the bulk value at ∼1.33 × 10${}^{-10}$ m${}^{2}$s${}^{-1}$, and calculated $\Delta {\bar{\mu }}_{Na,m}$, relative to $n_{f}=-2.5$. The diffusivity is expected to be smaller within a confined pore due to the nature of the binding sites [32,33] and, although this value may appear small, it produces a barrier-less conduction rate through the pore of ∼0.9 ×$10^{8}$ ions per second which is of the order of tens of pA. We choose $n_{f}=-2.5$ because the electrostatic contribution to add a third ion is zero, that is, E(3)−E(2)=0.

Both data sets are in excellent agreement with the theory, with currents only starting to differ at relatively large voltages when the experimental data deviate from Ohmic behaviour. Clearly beyond this regime, the system is far from equilibrium and our theory will need to be extended accordingly. After fitting we obtain $\Delta {\bar{\mu }}_{Na,1-4}$∼2.3,3.4,2.8,2.4kT when $n_{f}=-2.5$, with the sum of squared residuals being small at $10^{-4}$. When the concentration is 0.14M the ions face the following barriers to enter each site: ∼4.0,2.9,3.6,4.0kT. These barriers are fairly similar to each other, although it is clear that S2 is the more favoured site and this is shown by its occupancy. As already discussed and observed in Figure 6, the energy barrier at each site reduces when the bulk concentration increases from 0.14M, resulting in a larger ionic current. This is confirmed by predicted current-voltage dependencies for 0.25 and 0.5M solutions respectively as showing increases in current; and the current-concentration behaviour in Figure 7b. In this latter case the bulk solutions are assumed to be symmetrical, with the driving force originating from a 50 mV voltage drop. This curve clearly demonstrates increasing conduction with concentration and we note that the current is relatively small <10 pA and is continuing to increase even at 2M because the overall energy barrier to enter the pore is large. We expect that these predictions can be further refined if more experimental measurements can be made.

### 4.2. Comparison to Whole Cell Data

The theory can now be compared to the experimental whole-cell current-voltage recordings outlined earlier. In this experiment the data are normalised against the maximal current which is calculated when −10 mV is applied across the pore, and the bath solution contains 0.14M of Na${}^{+}$ ions. We note that in Figure 8a this normalisation is with respect to the absolute value of this maximal value.

Under experimental conditions only the bath solution was varied. As a result, the theoretical equilibrium concentration and (chemical potential) used to calculate the conductivity σ and hence current varies slightly at each experimental point. This is because they are defined from the average concentration (or chemical potential) from both bulk solutions. Since the chemical gradient is calculated from the difference in bulk concentrations, we consider the lower limit of bulk concentration to be 0.1 mM rather than 0, to avoid the gradient diverging at low concentrations. Even at with the lowest concentration being 0.1 mM, the gradient is ∼5kT and so at the edge of applicability of our theory.

In Figure 8a, we plot the normalised current-voltage curves for the range of bath solutions. Overall we see good agreement between theory and data, but with two exceptions. NaChBac is a voltage-gated channel so that, at negative voltages, the number of open channels is reduced because the open probability decreases resulting in a smaller overall current [5,34]. Thus, at voltages below −10 mV our current diverges from the experimental data, and hence serves as a prediction of the normalised current in a single open channel. This prediction is given by the dashed lines, which we note increase in magnitude as voltage becomes more negative because the gradient increases. Furthermore, when the bath solution contains no Na${}^{+}$ (black dashed curve) we observe poor agreement between theory and experiment and so highlight the curve with a dashed line. Finally, the inset curve shows the current closest to equilibrium.

The system is in equilibrium when the net current is zero, and this occurs when the applied voltage is equal to the reversal potential $\phi^{Re}$. This was measured experimentally and is compared to the theoretical current in (b). In the theory the reversal potential is calculated from,
(9)$$e\phi^{Re}=kT\log\left(x^{L}/x^{R}\right),$$
where L,R again refer to the left and right pipette/bath solutions respectively. We see good agreement except when the bath solution contains no Na${}^{+}$. Even, our reduced concentration of 0.1 mM yields a reversal potential smaller than −35 mV. This is echoed by the current at this concentration which is not in good agreement with the experiment (see the black dashed curve in Figure 8a). A possible explanation for these disagreements is that, in the absence of Na${}^{+}$ in the bath solution, K${}^{+}$ ions enter the pore but do not conduct, consequently blocking the pore. Furthermore, at this concentration we are at the limits of applicability because the chemical gradient is still relatively large ∼5kT. We plan to discuss this in a future manuscript after further investigations.

In Figure 9a we estimate the effective open probability $P_{\mathrm{eff}}$. This is defined relative to the open probability at peak current $P_{\max}$, from the ratio of theoretical and experimental current for each of the given concentrations. We neglect the estimate in the absence of Na${}^{+}$ because the theoretical current did not agree with experimental data. We observe that $P_{\mathrm{eff}}$ takes values between 0 and 1.5 except for three concentrations all at +50 mV of applied voltage. At 0.126M, 0.1386M and 0.14M bath concentration the theoretical current was below the experimental values and in the latter two concentrations of different sign. This produced estimated effective open probabilities, $P_{\mathrm{eff}}$, taking the values of 2.5, −15 and −0.5 for the three concentrations respectively (only $P_{\mathrm{eff}}∼-0.5$ is shown). Apart from these points however we observe it to be broadly sigmoidal and being 0 at negative voltages as anticipated. We expect, that the actual open probability, $P_{\mathrm{Open}}$, can be calculated through the following definition,
(10)$$P_{\mathrm{Open}}\left(V\right)=P_{\mathrm{eff}}\times P_{\max},$$
if the open probability of the maximal current is known.

In Figure 9b we highlight the current-concentration (I−C) behaviour by plotting the I−C curve at the peak voltage (−10 mV). Note that, unlike Figure 8a, the current is normalised to the maximum current at 0.14M (and not to the absolute value). As expected the theoretical current agrees fairly well with the experimental one except at low concentrations (≲5 mM). The curve takes a quasi-linear shape because the current comprises two terms: (1) the conductivity prefactor and (2) the electrochemical gradient. The second term is of the standard form, but our conductivity is a function of the equilibrium bulk chemical potential, which through our derivation must take the averaged concentration between the two bulks and thus slightly varies with bath concentration as well.

## 5. Conclusions and Summary

In summary, we have taken the statistical and linear response theory, originally derived in [28] and applied to KcsA and a mutant, and applied it to investigate Na${}^{+}$ conduction in NaChBac. Importantly, in order to compare with experimental and simulation data see Figure 2 and Figure 3), we needed to extend the theory to take account of a chemical gradient. In doing so, we derived the conductivity at each site and the total through the pore in the presence of an electrochemical gradient. The main result of the paper is the quantitative predictions of pore function that we make as a function of the energy profile, experimental bulk conditions, and the pore structure.

In Figure 7 we compared the theoretical current-voltage and equilibrium site occupancies to experimental and simulation data. This comparison allowed us to extract the following values of $\Delta {\bar{\mu }}_{Na,1-4}∼2.3,3.4,2.8,2.4kT$. At the experimental concentration 0.14M, the 3rd ion faces an energy barrier to enter each site within the pore of ∼4, 2.9, 3.6, 4kT. Although these values are not barrier-less as observed in KcsA [28], they are not expected to be because the experimental current is smaller in NaChBac. Furthermore, these parameters lead to barrier heights consistent with [8,20]. Using these parameters we have predicted the current for higher concentrations, including the current-concentration behaviour with 50 mV of applied voltage and current-voltage dependencies for 0.25 and 0.5M solutions. As expected both show an increase of current as the bulk solution increases. We expect that with more experimental data, we could refined these parameters.

In Figure 8 and Figure 9 we compared the theory to normalised whole-cell data, under the assumption that the normalisation effectively renders it a single-channel for the point of comparison. The theory was found to be in good agreement with experiment except for when the bath solution was devoid of Na${}^{+}$. A possible explanation is that in the absence of Na${}^{+}$, K${}^{+}$ ions enter the pore but do not conduct, subsequently blocking the pore. Furthermore, at this concentration we are at the limits of applicability because the chemical gradient is still relatively large ∼5kT. We plan to investigate this in a future manuscript by introducing a far-from equilibrium kinetic model that accounts for both Na${}^{+}$ and K${}^{+}$ ions. Such a model was briefly introduced in [20]. However, it failed to account properly for the correlations between ions at different sites, and only considered a 2 site pore; and so further development is needed.

Finally, we expect our theory to be applicable to the study of mixed-valence, that is, Na${}^{+}$/Ca${}^{++}$ selectivity in NaChBac and related voltage gated Ca${}^{++}$ channels, alongside artificial nano-pores.

## Figures and Tables

**Figure 1 entropy-23-00249-f001:**
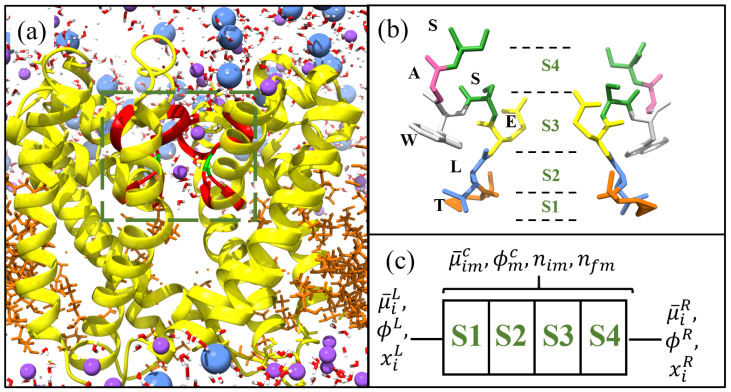
Structure of NaChBac [8] visualised using chimera [11]. (**a**) Yellow ribbons denote the protein spanning a lipid membrane (orange strands) between two aqueous ionic solutions. The selectivity filter (SF) is located within the box and highlighted by the red ribbons. The charged glutamates in the SF are highlighted green, and Na${}^{+}$ (purple), and Cl${}^{-}$ (blue) ions alongside water molecules are included. (**b**) Structure of the SF for NaChBac with each amino acid highlighted and labelled by colour. The positions of the binding sites are included and labelled S1–S4 from the intra- to the extra-cellular side respectively. In (**c**) we show the lattice model used to define the system.

**Figure 2 entropy-23-00249-f002:**
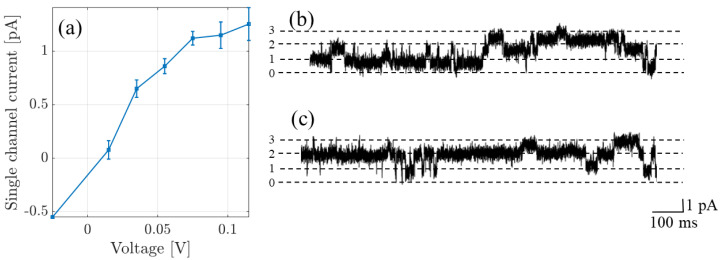
(**a**) Single channel currents recorded from NaChBac (originally published in [20]). (**b**,**c**) The original recording made at +100 mV in the 140 mM NaCl solution; the trace contains contributions from at least three active channels; and (**c**) represents a continuation in time of trace (**b**). The dashed lines show the amplitude level per channel, the numbers on the ordinate denoting the number of open channels.

**Figure 3 entropy-23-00249-f003:**
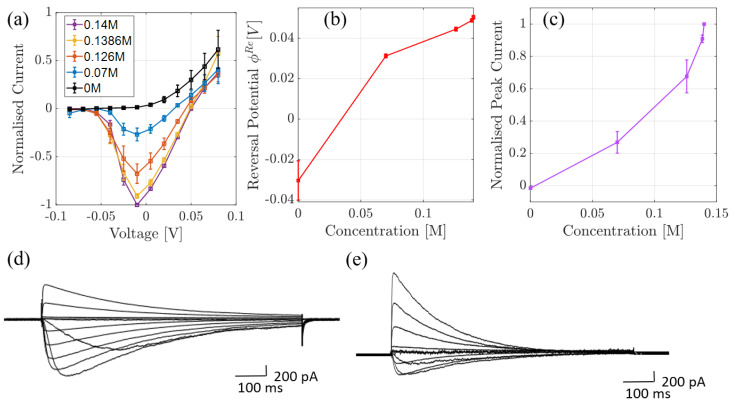
(**a**) Mean peak whole cell voltage-current relationships from cells expressing NaChBac channels, obtained in the bath solution with decreasing Na${}^{+}$ content ranging from 140 mM to 0 mM (with NaCl being replaced with equimolar KCl). The peak currents were determined from time vs. current traces (examples shown in parts (**d**,**e**). Peak currents are normalized to the peak current recorded from the same cell in 140 mM NaCl-containing solution in the absence of K${}^{+}$; error bars represent the standard error of the mean (SEM), determined from at least 4 independent cells. In (**b**) we show mean reversal potentials (±SEM) determined from data plotted in part (**a**). In cases where inward current was not detected, the reversal potential was assumed to be the voltage at which outward current could be detected. In (**c**) we plot the mean (±SEM) peak whole cell current (determined from data plotted in part a) as a function of Na concentration. Parts (**d**,**e**) are examples of time-dependent NaChBac currents recorded in 140 mM NaCl (**d**) and 126 mM NaCl and 14 mM KCl (**e**).

**Figure 4 entropy-23-00249-f004:**
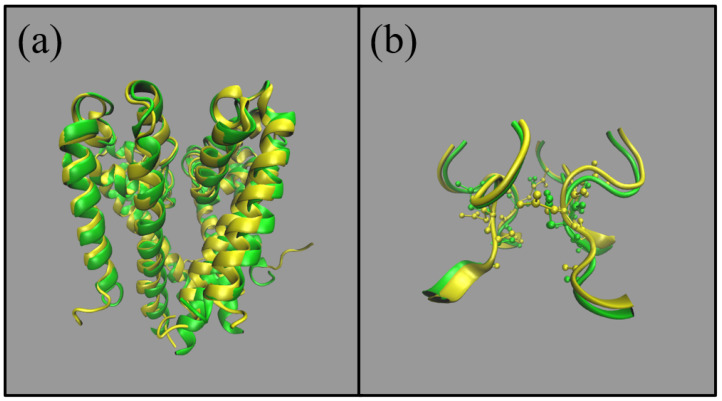
Comparison of NaChBac structures from the homology model (yellow) introduced in [8] and the Cryo-Em structure in green (6vx3.pdb) from [4]. (**a**) represents the whole pore and (**b**) is a snapshot of the (half) selectivity filter.

**Figure 5 entropy-23-00249-f005:**
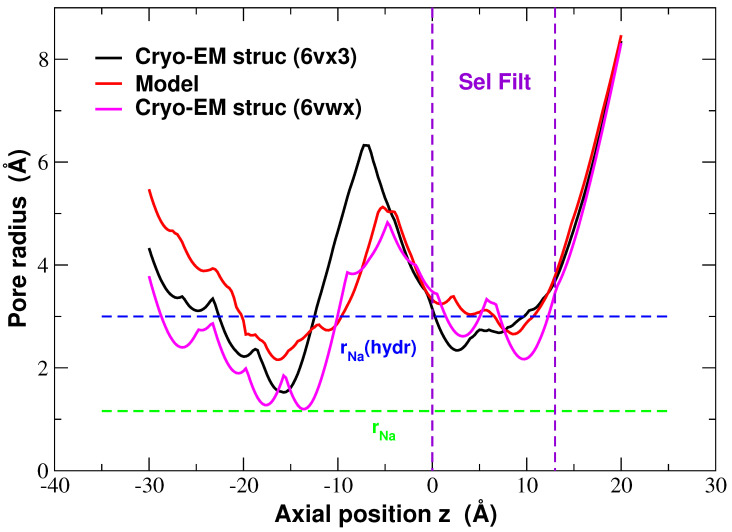
Comparison of average pore radius in the homology model structure (red) [8] and Cryo-EM structures 6vx3.pdb (black) and 6vwx.pdb (pink) [4]. The green and blue dashed lines denote the ionic Na${}^{+}$ and hydrated Na${}^{+}$ radii, respectively, and the purple dashed lines at z=0,13 Å highlight the selectivity filter region.

**Figure 6 entropy-23-00249-f006:**
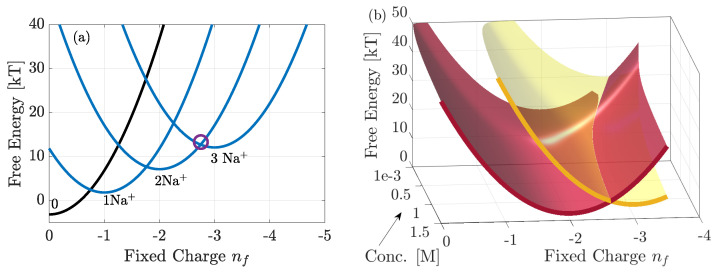
Free energy of the favoured states, plotted with $\Delta {\bar{\mu }}_{Na,1-4}$
∼2.3,3.4,2.8,2.4kT. In (**a**) it is plotted vs. $n_{f}$ with 0.14M NaCl bulk solutions and in (**b**) vs. both $n_{f}$ and bulk concentration. In (**a**) the blue curves correspond to the occupied n>0 states of the pore, and black denotes the empty state. The purple circle highlights the location at which the two most favoured 2 and 3 ion states coincide, and we see that at $n_{f}=-2.5$ there is a small energy barrier. As bulk concentration increases this energy barrier reduces and the purple circle would shift towards $n_{f}=-2.5$. This is further clarified by (**b**) which shows only the 2 and 3 ions states.

**Figure 7 entropy-23-00249-f007:**
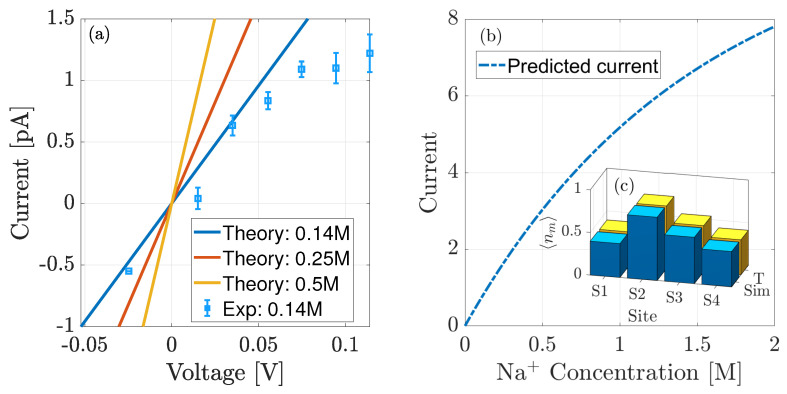
(**a**) Comparison of theoretical current vs. experimental data (squares) taken from [20] with symmetrical 0.14M NaCl solutions. (**b**) Predicted current-concentration curve at 50 mV across the pore. (**c**) Comparison of equilibrium occupancy at each site vs. simulation data with 0.5M NaCl solutions [8]. In doing this fitting we find that $\Delta {\bar{\mu }}_{Na,1-4}$
∼2.3,3.4,2.8,2.4kT, corresponding to energy barriers of ∼4,2.9,3.6,4kT at 0.14M and we find the pore diffusivity to be ∼1.33 × 10${}^{-10}$m${}^{2}$s${}^{-1}$.

**Figure 8 entropy-23-00249-f008:**
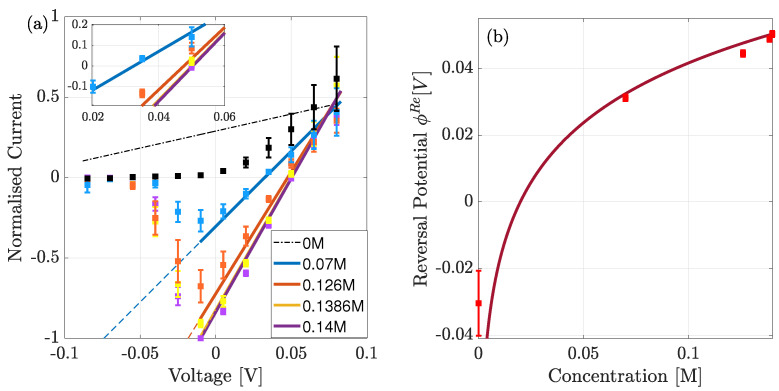
(**a**) Comparison of theoretical (solid line) to experimental (squares) data of normalised (to absolute value) whole-cell current in the presence of an electrochemical gradient, for a range of extra-cellular bulk solutions. The peak occurs at −10 mV, and below this voltage the current reduces due to the reduction in the open probability. Dashed lines predict the normalised currents if the open probability remained unchanged from the value at the peak current. (**b**) Theoretical (solid) and experimental (squares) of the reversal potential ($\phi^{Re}$) for a range of concentrations. Theory only differs when the right bulk is absent of Na${}^{+}$.

**Figure 9 entropy-23-00249-f009:**
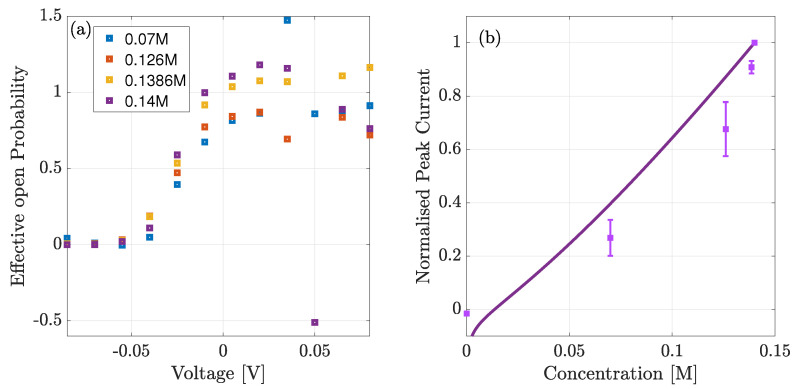
(**a**) Estimated open probability from the ratio of experimental to theoretical current. Below −40 mV the open probability is close to zero indicating that the channels are closed. (**b**) Comparison of normalised theoretical current (solid line) and experimental (squares) data vs. bulk concentration, at −10 mV of applied voltage.

**Table 1 entropy-23-00249-t001:** Table of averaged radii and length of each binding site, obtained through the homology based structural model of NaChBac from [8]. The corresponding surface areas and volumes were estimated by assuming that each site was spheroidal in shape. The binding site is identified from a minima in the potential of mean force (PMF), and its length is estimated from the distance between maxima in the PMF. The radius is estimated from the average calculated radius in this region. These lengths and radii are given in the table.

Site	Estimated Average Radius	Estimated Length	Estimated Surface Area	Estimated Volume
S1	3.06 Å	3 Å	116 (Å)${}^{2}$	117 (Å)${}^{3}$
S2	2.77 Å	4 Å	126 (Å)${}^{2}$	129 (Å)${}^{3}$
S3	2.75 Å	3 Å	90 (Å)${}^{2}$	80 (Å)${}^{3}$
S4	2.77 Å	2 Å	78 (Å)${}^{2}$	63 (Å)${}^{3}$
Mean	2.8 Å	3 Å	103 (Å)${}^{2}$	97 (Å)${}^{3}$

## Data Availability

Data related to this research are openly available from the University of Lancaster Research Directory at (https://doi.org/10.17635/lancaster/researchdata/421 (accessed on 17 February 2021)). Gibby, W.A.T.; Fedorenko, O.A.; Guardiani, C.; Barabash, M.L.; Mumby, T.; Roberts, S.K.; Luchinsky, D.G.; McClintock, P.V.E. (2020) Data for Application of a Statistical and Linear Response Theory to Multi-Ion Na${}^{+}$ Conduction in NaChBac [Dataset].

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
