# Peer review of "Application of a Statistical and Linear Response Theory to Multi-Ion Na+ Conduction in NaChBac"

_entropy, 2021, doi:10.3390/e23020249_

Round 1

Reviewer 1 Report

Gibby et al

This paper describes a biophysical study, involving modelling of transport
through passive ion channels. The choice of Entropy as a journal for this work
seems a little surprising, because the novel contributions are not related
to thermodynamics or information theory.

The paper describes (in the introduction) a structural model for the ion
channel, which is simplified to an array of four binding sites.
The conduction is then described in terms of a mathematical model for
this simplified system, as discussed in section 3.

The paper relies on references to other works, frequently paper with
overlapping authorship, particularly references [5,18,19]. I looked a [19]
on arXiv, and found that it uses essentially the same biophysical model,
and that the theoretical treatment is very similar, although a superficial
reading suggests that the version in [19] is more carefully presented.
While figures 1, 2 and 4 of that work are distinct from figures 1, 4 and 5
of this work, they convey the same ideas. There are also many similarities
with [18]. There are differences in the data presented here, but the differences
in the conceptual content appear to be minor, and not clearly highlighted.

Because of its close connection to earlier works by the same group of
authors, this paper is not really significant scientifically.
It adopts a detailed modelling approach which introduces a
great many parameters, and there is no deep insight, but the
paper is of comparable standard to much that does get published in the
biophysics literature. The paper is free from obvious errors and has been
carefully written and proof-read.

RECOMMENDATION. Primarily because of the substantial overlap with
references [18,19], I regard this work as being marginal as to whether
it should be published. If the editor of Entropy regards the subject
matter and the degree of overlap with [18] and [19] as acceptable,
the paper could be published with minor changes. Otherwise
I recommend a decisive rejection, because repeated review of
marginal-quality work is a poor use of referee's time.

Here are some points which might be addressed if the paper is going to be accepted.

1. Second paragraph: I couldn't understand the statement about a permeability
ratio of 0.04 being 'in closer agreement' to 170:1.

2. Third paragraph: presumably TLESWASS refers to the single-letter aminoacid code?
Remember this is a physics journal. Three letter codes are used elsewhere.

3. Page 2. The biophysical model of the channel is based upon homology
studies, rather than direct experiments. Homology with what?

4. There must be substantial uncertainty about the presumed structure, as
well as arbitrariness about how the pore dimensions are measured. How are the
dimensions listed in table 1 determined.

5. p.2 line 60: '...single-ion potential of mean force...'. Does not make sense
to me.

6. Exactly what do the 'red ribbons' highlighting the the selectivity filter
in figure 1 show?

7. Legend of figure 2 is confusing: panels (a)-(c) appear to refer to steady
current experiments (not clearly explained), whereas (d), (e) show transients,
also not explained.

8. There were several places in the text of section 3 where I had to look ahead
to find definitions of symbols.

9. Section 4: I didn't see where the data providing the quantitative information
in figure 4 comes from. What is ICB theory?

10. Section 4.1: the text doesn't explain how the 'fitting' was performed to
extract the potentials \Delta \mu=2.3,3.4,... etc. Which data was used, and
why should we believe that these fits are well-conditioned?

Reviewer 2 Report

The manuscript by Gibby et al. reports on an extremely interesting investigation of the conduction properties of the sodium-selective prokaryotic channel NaChBac. The work builds on a theory recently introduced by the authors and further developed here. I find the theoretical approach of the authors to be insightful and the manuscript instructive. The comparison with experiments and molecular simulations is compelling. I have only a couple of minor concerns: 

1) the authors fit the excess chemical potential of each site to reproduce the occupancy calculated from molecular dynamics simulation. My question is: how was the configurational space partitioned to compute occupancies? Are binding sites discrete or is there any more sophisticated geometric criterion to define the boundary between, let's say, S1 and S2? How much do the final results depend on these choices?

2) It is true that "Unfortunately, a crystal structure of NaChBac is not available.", however a high resolution Cryo-EM structure is available (Gao et. al ). This should be acknowledged in the text. Most importantly, a structural comparison between this experimental structure and the comparative homology model would provide confidence in the fact that the structure of the selectivity filter has been accurately predicted and therefore the simulations are reliable.

Minor: typo in the first sentence of the abstract: "Biological ion channels are of fundamental to maintaining life". 

Round 2

Reviewer 1 Report

The authors have made a careful response to the points raised by the 
referees, and the manuscript has been improved substantially. I recommend 
publication in Entropy.